Atlas of tissue-specific and tissue-preferential gene expression in ecologically and economically significant conifer Pinus sylvestris

Cervantes Sandra 1 2
Vuosku Jaana 1
Pyhäjärvi Tanja 1 3 tanja.pyhajarvi@helsinki.fi
1 Department of Ecology and Genetics, University of Oulu , Oulu , Finland
2 Biocenter Oulu, University of Oulu , Oulu , Finland
3 Department of Forest Sciences, Faculty of Agriculture and Forestry, University of Helsinki , Helsinki , Finland
Wink Michael
Electronic publication date: 2021 Aug 18
Publication date: 2021
Volume: 9
Electronic Location ID: e11781
Received 2021 Jan 13; Accepted 2021 Jun 24
Copyright: © 2021 Cervantes et al.
Copyright year: 2021
Copyright holder: Cervantes et al.
License: This is an open access article distributed under the terms of the Creative Commons Attribution License, which permits unrestricted use, distribution, reproduction and adaptation in any medium and for any purpose provided that it is properly attributed. For attribution, the original author(s), title, publication source (PeerJ) and either DOI or URL of the article must be cited.
License URL: https://creativecommons.org/licenses/by/4.0/

Keywords: Pinus sylvestris, RNA-seq, Tissue-specific gene expression, Conifer, Transcriptomics, Megagametophyte, Needle, Bud, Embryo, Phloem

Funding: Academy of Finland 287431, 293819, 319313 and 329438 Biocenter Oulu GenTree 676876 This work was supported by the Academy of Finland (grants 287431, 293819, 319313 and 329438) and Biocenter Oulu. This publication is part of a project that has received funding from the European Union’s Horizon 2020 research and innovation programme under grant agreement No. 676876 (GenTree). The funders had no role in study design, data collection and analysis, decision to publish, or preparation of the manuscript.

==============================
Despite their ecological and economical importance, conifers genomic resources are limited, mainly due to the large size and complexity of their genomes. Additionally, the available genomic resources lack complete structural and functional annotation. Transcriptomic resources have been commonly used to compensate for these deficiencies, though for most conifer species they are limited to a small number of tissues, or capture only a fraction of the genes present in the genome. Here we provide an atlas of gene expression patterns for conifer Pinus sylvestris across five tissues: embryo, megagametophyte, needle, phloem and vegetative bud. We used a wide range of tissues and focused our analyses on the expression profiles of genes at tissue level. We provide comprehensive information of the per-tissue normalized expression level, indication of tissue preferential upregulation and tissue-specificity of expression. We identified a total of 48,001 tissue preferentially upregulated and tissue specifically expressed genes, of which 28% have annotation in the Swiss-Prot database. Even though most of the putative genes identified do not have functional information in current biological databases, the tissue-specific patterns discovered provide valuable information about their potential functions for further studies, as for example in the areas of plant physiology, population genetics and genomics in general. As we provide information on tissue specificity at both diploid and haploid life stages, our data will also contribute to the understanding of evolutionary rates of different tissue types and ploidy levels.

Introduction

Conifers, a clade within the gymnosperms, represent a group of plants with significant economic and ecological relevance (Farjon, 2008). Several coniferous trees, for example in Pinus and Picea genera, are among the most important sources of wood and timber (San-Miguel-Ayanz et al., 2016; Food and Agriculture Organization of the United Nations, 2020). Conifers dominate boreal forests worldwide and can form large forested areas hosting a variety of ecosystems. Furthermore, conifer forests are one of the major ecosystem services providers and they are crucial for carbon sequestration (Bonan, Chapin & Thompson, 1995; DeAngelis, 2008; San-Miguel-Ayanz et al., 2016; Boonstra et al., 2016). Despite their importance, genomic resources for conifers, and gymnosperms in general, lag behind in availability compared to angiosperms. Although several contributions have been made recently to fill this gap (Nystedt et al., 2013; Birol et al., 2013; Zimin et al., 2014; Stevens et al., 2016; Mosca et al., 2019), conifer genome annotation remains a challenge, with both structural and functional annotations being far from perfect (Wegrzyn et al., 2014; Cañas et al., 2019). Conifer genomics resources are limited due to the large size of their genomes, ranging from 8 to 70 Gbp (Zonneveld, 2012) and to the large number of repetitive elements (approximately 80%) within them (Nystedt et al., 2013; Neale et al., 2014; De La Torre et al., 2020). Proper and complete annotation of the conifer genomes has also been complicated by the presence of long introns (Nystedt et al., 2013; Wegrzyn et al., 2014), which prevents the routine use of common annotation software. Moreover, analyses of ortholog genes across different species indicate that there are several gene groups which are unique to conifers or conifer species specific, with no well-defined homologs in any of the angiosperm plant models (Nystedt et al., 2013; Wegrzyn et al., 2014; Neale et al., 2014; Baker et al., 2018).

Transcriptomic resources have been particularly important for research in conifers and other non-model species, as a strategy to compensate for the challenges associated with efficient genome assembly and annotation (Cañas et al., 2019; Wegrzyn et al., 2020). As the biological functions can not be directly inferred from nucleotide sequences, reference transcriptomes and gene expression studies are useful in the identification and annotation of genes (Raherison et al., 2012; Wegrzyn et al., 2014; Merino et al., 2016; Little et al., 2016; Cañas et al., 2017). Transcriptome information can also be used in conifers that lack reference genomes, as this information can be used in the design of reduced genome representation targets (Rellstab et al., 2019; Tyrmi et al., 2020). In addition to this, RNA-seq analyses allow the identification of expression patterns and expression levels, which are essential components of evolutionary genomics studies. For example, selective constraints in genes can be inferred from their expression patterns, as both breadth and expression level are known determinants of evolutionary rates (Wright et al., 2004; Slotte et al., 2011). Selective constraints are also expected to differ between haploid and diploid tissues which differ in the relative rate of expression, as tissue specificity and ploidy has potentially drastic effects on the dynamics of for example, purifying selection (Otto, Scott & Immler, 2015).

Here we give a first glimpse of the expression patterns of tissue preferentially upregulated (PUR) and tissue specifically expressed genes across five organs (embryo, megagametophyte, needle, phloem and vegetative bud, hereafter called tissues, but see discussion) of Pinus sylvestris. P. sylvestris is a widely distributed conifer of large economic and ecological importance in Northern Eurasia (Pyhäjärvi, Kujala & Savolainen, 2020). P. sylvestris is one of the main sources of timber and raw material for the pulp and paper industry in Europe and is a dominant species in boreal forests, with an estimated coverage area of 145 millions hectares (Pyhäjärvi, Kujala & Savolainen, 2020). P. sylvestris is also a suitable model to answer evolutionary and genetic questions, especially regarding gymnosperm reproductive biology, its evolution and genetic consequences. For example, in conifers the maternal nuclear haplotype of an embryo is identical to the megagametophyte’s nuclear haplotype (Williams, 2008), which makes it possible to separate expression of paternal and maternal haplotypes and alleles in the embryo (Verta, Landry & MacKay, 2016).

Despite its importance and potential, P. sylvestris still lacks a reference genome, and currently there are limited genomic resources for this species (see however Wachowiak et al., 2015; Merino et al., 2016; Li et al., 2017; Höllbacher et al., 2017; Ojeda et al., 2019; Perry et al., 2020). To date, the few transcriptomic studies of P. sylvestris have been based on a small number of tissue types such as needles or seed tissues (Wachowiak et al., 2015; Merino et al., 2016). Identification of tissue preferentially upregulated and tissue specific genes is relevant because (1) understanding the patterns of expression across different kinds of tissues can aid to elucidate the organization of transcriptomes (Raherison et al., 2012). (2) Knowing the different profiles of expression across tissues can set the ground for evolutionary analysis, as it is known from studies in mammals and angiosperms that the evolution of gene expression differs across tissues and organs (Brawand et al., 2011; Yang & Wang, 2013). Ultimately this knowledge will help to gain a deeper understanding of the determinants and main factors that affect the rate of adaptive evolution and the dynamics at the genome level.

In this study we (1) provide a comparative transcriptomic resource for P. sylvestris describing the expression level in five different tissues, (2) identify genes that are tissue preferentially upregulated and tissue specifically expressed in each of the five tissues, (3) provide quantitative measures of tissue-specific expression for each gene per tissue combination, and (4) conduct gene ontology enrichment analysis for each tissue type. Our results are important for future studies in comparative conifer genomics, plant physiology, population genetic analyses, evolutionary genetic studies, further gene expression analyses, and aid in the annotation of present and forthcoming conifer genome sequences.

Materials and Methods

Plant material and RNA sequencing

We used the RNAseq data (trimmed reads, BioProject PRJNA531617) previously used to assemble multiple reference transcriptomes of P. sylvestris by Ojeda et al. (2019). The plant material was obtained during the growing season of 2016 (May 26th–27th) and consisted of needles, phloem, vegetative buds and seeds from six non-related adult Pinus sylvestris trees. We used data from trees growing in a natural forest population to ensure that the observed patterns of tissue preferential expression and specificity were robust to genotypic and environmental variation. The material was collected at the Punkaharju Intensively Studied Site (ISS) in Southern Finland (Table S1). Samples were collected in collaboration with Natural Forest Research Institute Finland LUKE that has an agreement on the forest research use with the owner Metsähallitus (The Finnish Forest Administration). In Finland P. sylvestris is not endangered and Finland does not regulate its genetic resources under Nagoya Protocol so CITES was not applied and prior informed consent was not needed. The samples were collected and identified by Tanja Pyhäjärvi. There is no voucher available for the specimens.

Relevant experimental procedures of Ojeda et al. (2019) are briefly summarized here. Megagametophyte and embryo tissues were obtained by dissecting mature seeds collected from the same mother trees from which the vegetative tissues were obtained. Seeds were stored in the dark at 4 °C until germination was induced by exposure to moisture and continuous light for 48 h. Total RNA was extracted from needle, bud and phloem using the Spectrum Plant Total RNA Kit (Protocol B; Sigma, Kawasaki, Kanagawa, Japan), followed by mRNA capture with the NEBNext® Poly(A) mRNA Magnetic Isolation Module (New England Biolabs Inc., Ipswich, MA, USA). For embryo and megagametophyte, mRNA was directly extracted from the whole tissues with Dynabeads mRNA Direct Micro Kit (Thermo Fisher Scientific, Waltham, MA, USA). RNA concentration was quantified with Qubit RNA HS Assay kit (Thermo Fisher Scientific, Waltham, MA, USA). The quality and integrity of the RNA was visually assessed with a 2100 Bioanalyzer using the RNA 6000 Pico kit (Agilent, Santa Clara, CA, USA). A total of 30 libraries were prepared by using the NEBNext Ultra Directional RNA Library Prep Kit for Illumina (New England Biolabs Inc., Ipswich, MA, USA). Sequencing (2 × 150 bp) was conducted with an Illumina NextSeq 500 at the Biocenter Oulu Sequencing Centre (Oulu, Finland).

Transcript quantification and abundance matrices construction

We followed the Trinity Post-Transcriptome Assembly Downstream Analyses pipeline (Trinity v.2.6.6) (Haas et al., 2013, https://github.com/trinityrnaseq/trinityrnaseq/wiki/Trinity-Transcript-Quantification) to generate quantification files at isoform level, and raw counts and normalized count matrices at putative-gene level (hereafter referred as gene level matrices). For transcript quantiﬁcation we used the trimmed reads from Ojeda et al. (2019) and as reference we used the Trinityguided transcriptome reported in the same work (Data S1). We chose the non-reduced redundancy Trinityguided transcriptome as reference (instead of the redundancy reduced assembly) to avoid mapping of reads from different paralogous genes to the same contig. P. sylvestris has a large amount of paralogous and repetitive regions, and with this we reduced the amount of false mapping across paralogs due to sequence similarity.

To obtain independent transcript abundance estimates of each of the six individuals in each of the five tissues we used Salmon 0.9.1 (Patro et al., 2017) as implemented in the Trinity pipeline, with the --SS_lib_type (strand specific) and --trinity_mode options. The –trinity_mode option generated a transcript-to-gene map that allowed the estimation of counts from isoforms to generate counts at a putative gene level during the count matrix generation step. Before any further analysis, we checked for the presence of possible contaminants by searching contigs that had hits to the keywords ‘alveolata’, ‘metazoa’, ‘fungi’, ‘bacteria’, and ‘archaea’. We search for exact matches to these keywords from the results of a translated blast (BLASTX) of the transcriptome annotation file (Ojeda et al., 2019; Ojeda, 2020). We then combined our list of putative contaminants with the contaminants and organelles contigs lists reported in Ojeda et al. (2019), and excluded them from the isoform quantification files and the gene_trans_map. Contaminants were removed after the transcript quantification stage to avoid the false mapping of contaminant reads to non-contaminant contigs in the reference transcriptome.

We built three count matrices at the gene level based on the clean independent transcript quantification estimates with the script abundance_estimates_to_matrix.pl from the Trinity pipeline. For this, we generated a gene level raw counts matrix (Table S2), which was then used to construct a transcript per million length normalized gene count matrix (TPM escalated matrix) (Table S3). The TPM escalated matrix accounts for differences in isoform lengths that otherwise could inflate FDR due to differential transcript usage (Soneson, Love & Robinson, 2015). Finally, the TPM escalated matrix was used to construct a gene counts matrix normalized using the Trimmed Mean of M values (TMM) method (Table S4), which accounts for differences in the distribution of transcript expression that could lead to an increase in false positive rates, and decrease the power to detect truly differentially expressed genes (Robinson & Oshlack, 2010). Before doing the differential expression analyses and the estimation of tissue specificity, we evaluated the quality of our samples by doing a principal component analysis (PCA) and a Pearson correlation matrix using the gene raw count matrix, according to the Trinity QC samples and biological replicates pipeline (https://github.com/trinityrnaseq/trinityrnaseq/wiki/QC-Samples-and-Biological-Replicates). The intention of these analyses was to look for the presence of batch effects or sample outliers, and to verify that biological replicates clustered within each tissue type and not among sampled individuals.

Differential expression analysis and identification of tissue preferentially upregulated genes

Differentially expressed genes (DEG) and preferentially up-regulated genes (PUR) were identified using the Trinity Differential Expression and Sample-Specific Expression pipelines (Bryant et al., 2017, https://github.com/trinityrnaseq/trinityrnaseq/wiki/Trinity-Differential-Expression). Briefly, we first identified DEG using the gene raw counts matrix with edgeR 3.28.0 (Robinson, McCarthy & Smyth, 2010; McCarthy, Chen & Smyth, 2012). The differential expression analysis was based on pairwise comparisons of each of the five tissues, using the six samples per tissue as biological replicates, then for each pair of DEG identified we obtained their associated false discovery rate (FDR). Next, we obtained a normalized mean value of expression for each tissue by averaging and log2 transforming the counts for each gene across the six replicates for each tissue on the TMM gene matrix. Afterwards, pairwise comparisons of the averaged log2 counts values per tissue were done and a logFC was assigned to each gene. DEG with a maximum FDR of 0.05 for differential expression, and with positive logFC in each pairwise comparison of the averaged log2 TMM normalized counts was then classified as PUR. A summary of pairwise expression differences between tissues based on the logFC of the log2 transformed gene counts in the TMM matrix is provided in Data S2.

Tissue-specific expression

As an alternative approach to quantitatively assess the tissue-specific expression of the genes we calculated the τ index based on the TMM gene counts matrix. The τ index ranges between 0 for widely expressed genes and 1 for exclusively tissue-specific genes (Yanai et al., 2005). As the τ index considers tissue specificity independently of the level of expression, we set as “not expressed” genes with expression values <1 from our TMM matrix in order to exclude genes with low support for true expression and low signal to noise ratio. To do this, we first log2 transformed the matrix in order to normalize the distribution of the expression values. We set all negative values in the matrix to zero, as this represented values <1 before log2 transformation. We excluded contigs that had no expression values or that had expression in just one out of the 30 samples. Then, the τ index was computed separately for each gene across all tissues and replicates following according to the following equation (Yanai et al., 2005; Kryuchkova-Mostacci & Robinson-Rechavi, 2017, https://github.com/severinEvo/gene_expression/blob/master/tau.R):

τ=∑i=1N⁡(1−Xi)N−1,Xi=ximax(xi)wheremax(xi)1≤i≤N

where N represents the number of tissues, xi is the mean expression in tissue i and Xi is the expression level in tissue i normalized by the maximum mean expression among all tissues.

Singular enrichment analysis

To further characterize the gene expression in the five tissues, we identified the biological pathways for both tissue-specific and tissue preferentially upregulated gene sets with independent singular enrichment analysis (SEA) (Huang, Sherman & Lempicki, 2009; Du et al., 2010). First, we retrieved the UniProt IDs corresponding to our putative genes from the blastx field from our reference annotation file (Ojeda et al., 2019). Then we uploaded the list of UniProt IDs to the uniprot retrieve/ID mapping tool (https://www.uniprot.org/uploadlists/) and restricted the result to GO terms only. We repeated this procedure with the genes used as a background list for the SEA: all the contigs in the gene raw counts matrix for the PUR genes (Data S3), and all the contigs in the filtered TMM matrix in the case of the tissue-specific genes (Data S4).

Of the 715,398 putative genes in the raw counts matrix used for the differential expression analysis, 17,227 have a unique UniProt ID and represent 108,947 GO terms. The background list for the tissue-specific genes data set consisted of 177,075 contigs of which 14,079 have a unique annotation and represent 90,198 GO terms. For both data sets only uniquely annotated genes and their corresponding GO terms (Data S5–S14) were used for running the singular enrichment analyses to avoid inflating the number of GO terms falsely, and creating a bias in the analysis.

We used the GO terms along the UniProt IDs as input for the SEA using the agriGO (http://systemsbiology.cau.edu.cn/agriGOv2/index.php) platform (Du et al., 2010; Tian et al., 2017). We used the custom background list option, applied a hypergeometric test as statistical test method with a minimum of five mapping entries per term, and Hochberg FDR as multi-test adjustment method with a significance level of 0.05. As the significant enrichment of child terms can inflate the enrichment significance of parental terms, after the SEA we used REVIGO (Supek et al., 2011, http://revigo.irb.hr/) to reduce the redundancy of the GO terms and highlight the unique and non-dispensable terms per tissue for both PUR and tissue-specific genes. We used the list of enriched GO terms found in the SEA and their respective p-value as input, selected the small output setting for redundancy, the whole UniProt as database and SimRel as measure of semantic similarity.

Results and discussion

Transcript quantification and abundance matrices construction

We mapped a total of 707,063,773 trimmed and adapter removed reads from five different tissues (embryo, megagametophyte, needle, phloem and vegetative bud) and six biological replicates (six different genotypes) per tissue type to P. sylvestris TRINITYguided transcriptome (Ojeda et al., 2019). On average 23,568,792 reads originated from each tissue, ranging from 29,591,629 reads for needle to 20,469,80 reads for phloem. On average 76% of the reads per replicate were successfully mapped to the reference (Table S5). After mapping 1,307,500 contigs had aligned reads at the isoform level. Of those, 119,882 contigs were removed from the downstream analyses as they were identified as contaminants (Data S15). The final set consisted of 1,187,460 contigs at isoform level and were used to construct raw counts and normalized matrices at gene level for downstream analyses (see Materials and Methods section). The total number of putative genes with expression signal in the gene level matrices was 715,398, much higher than the number of annotated genes in any conifer (Nystedt et al., 2013; Neale et al., 2014; Gonzalez-Ibeas et al., 2016). This magnitude, albeit probably an overestimate, is typical to transcriptome studies (Little et al., 2016). This is likely a result of single genes being present in multiple fragments, isoforms split into multiple genes, and different alleles originating from heterozygous material identified as separate genes during assembly and classification as genes by Trinity (Ojeda et al., 2019). However, part of the genes originate from gene families and since clustering similar genes is possible in downstream analysis, we chose to err on the side of potentially over splitting the genes rather than imperfectly clustering similar transcripts as a single gene, as over clustering will inherently lead to loss of information. We believe that providing expression data with minimum clustering will be most versatile for later use of the transcriptome and expression data in genome annotations and other studies.

Quality assessment of biological replicates

As we used different genotypes as biological replicates, we first verified that the replicates clustered by tissue type and not by genotype, and checked for the presence of potential outliers in the dataset. We used the raw counts matrix data (Table S2), a principal component analysis (PCA) and a Pearson correlation to verify this. The PCA separated the tissue samples into five distinct clusters without any overlap, indicating that among-tissue variation is the main factor of among-sample variation (Fig. 1). Hence, our approach captures the differentiating gene expression profiles of the five tissues. In the PCA, the seed-derived megagametophyte and embryo samples clustered closest to each other, suggesting similarity in their gene expression profiles. Also phloem and bud samples clustered close to each other, whereas needle samples showed the most unique gene expression profile. In the hierarchical clustering analysis, based on the correlations of gene expression profiles, the differences among tissues are relatively shallow. But, similarly to the PCA, all replicates are clustered according to their tissue type and not according to their genotypes, corroborating the PCA results (Fig. S1).

Figure 1 Schematic representation of the five tissues used and a scatterplot of the first two axes of the principal component analysis (PCA) of expression.

(A) Schematic representation of the five tissues used in the transcriptome profiling of Pinus sylvestris: needle, vegetative bud, megagametophyte, embryo and phloem. (B) Scatterplot of the first two axes of the principal component analysis (PCA). Tissue types are denoted by colors. Illustrations by Dorota Paczesniak.

Tissue preferentially upregulated and tissue-specific gene expression

We defined a gene as tissue PUR when there was a significant log fold change in the expression value compared to the other tissues. To identify tissue PUR we first did a differential expression (DE) analysis. For this we included all the genes in the raw count matrix (Table S2). We decided not to apply any minimum number of counts per gene as a filtering threshold to run the analysis, as we later applied a 5% false discovery rate (FDR) threshold for the identification of PUR genes. Out of the 715,398 genes initially included in the DE analysis, 198,413 genes had a maximum 5% FDR for differential expression and were further included in the analysis to identify PUR genes. We identified a total of 48,001 genes with tissue preferential expression, and out of the five tissues needle has the highest number of PUR genes (Table 1)

Table 1 Genes identified as preferentially upregulated and tissue specific in five P. sylvestris tissues. The percentage of unique UniProtKB identifiers is also shown.

	Tissue preferentially upregulated genes	Tissue specifically expressed genes	
Total	Annotated	Unique (%)	Total	Annotated	Unique (%)	
Bud	8,225	2,515	30.6	693	342	49.3	
Embryo	10,430	2,820	27.0	498	206	41.3	
Megagametophyte	7,171	1,515	21.1	679	220	32.4	
Needle	13,128	3,993	30.4	1,495	603	40.3	
Phloem	9,047	2,603	28.7	534	202	37.8	

Quantification of tissue specificity allows a powerful statistical analysis of correlation between tissue-specific expression and for example, evolutionary rate or other dependent or explanatory variables and factors. We identified the tissue specifically expressed genes by calculating the τ score per gene. The score ranges from zero to one, with a zero given to genes expressed in all tissues and one given to completely tissue specific genes. For this analysis we retained a set of 177,075 genes (Table S6) after applying the filtering criteria described in Methods. We considered a gene as tissue specifically expressed only if its τ = 1. We identified a total of 3,899 genes with a tissue-specific pattern of expression. Similarly, the PUR analysis results, needle has the highest number of tissue-specific genes (Table 1). To obtain the annotation of the genes identified as tissue PUR and tissue specific, we retrieved the corresponding UniProtKB identifiers (Ojeda, 2020) from the Trinotate for the 715,398 putative genes in the TMM count matrix, out of which 97,435 (14%) had a Swiss-Prot (Bairoch & Apweiler, 2000) protein match based on BLASTX (Ojeda et al., 2019). Most of the Swiss-Prot annotations (67%) originated from Arabidopsis thaliana (65,214 genes). Other common annotation sources were Nicotiana tabacum (9,794; 10%) and Oryza sativa (8,946; 9%). Only 1,663 genes (1.7%) had an annotation to other Pinus species, of which 177 (10.6%) were hits to P. sylvestris, and 608 (36.5%) genes had Swiss-Prot annotation to Picea. Note that Swiss-Prot is a manually curated database that does not currently have a comprehensive set of annotated gymnosperm proteins and therefore the best matches are often obtained from the model plants such as A. thaliana. A proportion of our putative genes share the same gene identifier (annotation) (Table 1). This probably reflects the incomplete collapse of different isoforms in the assembled transcriptome used as reference, or the presence of gene families (Wegrzyn et al., 2014). Also, a high number of the genes identified as PUR or tissue specific lack annotation altogether, which is not surprising as genes with higher tissue-specific expression have less conserved sequences and are less likely to find orthologs among other species (Lemos et al., 2005; Raherison et al., 2012). A summary of the 715,398 genes indicating their normalized expression level (TMM), τ score, tissue specificity status, PUR status, and annotation can be found in the Supplementary information (Table S7).

Cursory inspection of annotations of highly expressed tissue PUR and tissue-specific genes are congruent with some of the already known functions of the tissues. These results confirm that our analyses capture biologically meaningful characteristics of the tissues. For example in megagametophytes, enzymes related to seed storage lipid mobilization and germination were upregulated and specifically expressed. Similarly, in needles, several chlorophyll a–b binding proteins are upregulated. In embryo, multiple ribosomal proteins and other proteins indicating active protein synthesis were upregulated. In vegetative buds, expression of genes involved in defense against insect attack, like (−)-alpha-pinene synthase and dirigent (Ralph et al., 2006) that take part in oleoresin synthesis, were highly expressed and specific to this tissue. In phloem, the two genes annotated as metallothionein-like protein EMB30, an aquaporin and a thioredoxin-like protein were highly expressed, similarly to Quercus suber phellem (cork) where metallothionein reacts to oxidative stress (Mir et al., 2004) or in Pinus taeda xylem where the same proteins were among the most highly expressed genes (Lorenz & Dean, 2002).

Among the five tissues analysed, the needle had the highest number of genes with tissue-specific expression and embryo the lowest (Table 1). Except for two genes, one in megagametophyte and one in needle, all the genes with tissue-specific expression were also among the PUR genes. However, as tissue specificity does not require a high expression level, genes with τ score equal to one are not necessarily the most upregulated genes in their respective tissues. Comparison of our findings to other studies is not straightforward as there are very few transcriptomic studies in P. sylvestris. But in comparison to a previous study (Merino et al., 2016), where they focus on the comparison between megagametophyte and embryo tissues at different developmental stages, we identified less megagametophyte and embryo specifically expressed genes. One of the reasons for this difference could be that the identification of unique genes in Merino et al. (2016) was based only on the comparison between embryo and megagametophyte tissues. As the identification of tissue specific genes is contingent to the number of tissues used for the analysis, it is expected that the higher the number of tissues used in the comparison, the lower the number of tissue specific genes that will be identified. In contrast, we found a higher number of tissue specific genes in embryo, bud and needle compared to a previous study in conifers (Raherison et al., 2012), where several tissue types were used. One notable difference between this (Raherison et al., 2012) and ours was the higher number of tissue-specific genes for megagametophyte found in P. glauca. Raherison et al. (2012) found the highest number of unique genes in the megagametophyte in comparison to other tissues analyzed. The low number of megagametophyte specific genes identified in our study could be due to the use of mature embryos as starting material as previous research suggests that the number of unique transcripts in the megagametophyte varies during the developmental stages of embryogenesis (Merino et al., 2016).

One caveat of our analyses is that, unlike other studies, we did not use microdissection in order to obtain the tissue samples (Cañas et al., 2017). Hence, some of the “tissues” are a mix of tissue types. Needles, for example, include several tissues (phloem among them) (Pongrac et al., 2019), and mature embryos contain the shoot and root meristems as well as cotyledons (Singh, 1978). In contrast, the mature megagametophyte is a quite uniform storage tissue consisting of cells packed with starch protein and lipids (Simola, 1974; Vuosku et al., 2015). Another limitation of the dataset is that it represents only one point in time and space, although gene expression is a dynamic process and quantitative and qualitative variations exist over spatial and temporal scales. Instead of sampling across several developmental stages or across a spatial gradient our dataset represents a wider set of tissues, which increases the power to identify tissue PUR and tissue specifically expressed genes. The added value of the dataset lies in the unexpected functions and connections discovered among biological pathways and genes with previously unidentified signals of tissue-specificity or up-regulation.

Functional characterization of tissue preferentially upregulated and tissue-specific genes

GO enrichment analysis allows the identification of gene functions enriched with certain functional roles. The number of enriched functions was of the same magnitude across tissue types, ranging from 253 to 452 for PUR genes and from 58 to 169 for tissue-specific genes (Tables S8–S17). The total number of GO terms, the number of significant enriched terms, and the number of terms after the reduction of redundancy are shown in Table 2. Most of the genes (86%) with expression signals in our study lacked annotation from the Trinotate pipeline. Thus, they did not contribute to functional analysis or GO enrichment results. A summary of the most highly expressed genes per tissue, and the most enriched, non-redundant GO terms in the biological processes category are shown in Fig. 2. The complete lists of gene identifiers and their corresponding GO terms per tissue and per each set of genes (Data S5–S14), along with tables with the results of the SEA showing each GO terms, its p-value, and FDR (Tables S8–S17), and lists with levels of uniqueness or indispensability for each significantly enriched term in the five tissues (Data S16) are provided in supplementary information.

Figure 2 Ten most significant non-redundant biological processes.

Ten most significant non-redundant biological processes and their corresponding GO-term IDs (terms chosen based on the lowest dispensability value), and ten most highly expressed annotated genes in each of the five tissues. Genes preferentially upregulated (PUR) in a given tissue are in bold. Illustration and design by Dorota Paczesniak.

Table 2 Number of significantly enriched GO terms, and number of non-redundant terms in P. sylvestris tissues.

	Tissue preferentially upregulated genes	Tissue-specific genes	
Total	Significant	Non-redundant	Total	Significant	Non-redundant	
Bud	15,681	452	169	2,019	137	65	
Embryo	17,461	253	182	1,178	75	50	
Megagametophyte	9,690	306	123	1,363	111	51	
Needle	25,295	401	170	3,818	169	81	
Phloem	16,371	422	181	1,249	58	40	

In needles the significant GO terms reflected the exposure of trees to various stresses and interactions with other organisms, whereas in embryos, buds and the phloem the GO terms were mainly connected to different development-related processes. In needles the enriched biological process GO terms among tissue-specific genes were related to immune response (GO:0006955) as well as response to stress (GO:0006950) and other organisms (GO:0051707) such as oomycetes (GO:0002229), bacteria (GO:0042742) and fungi (GO:0009817). Moreover, terpene synthase activity (GO:0010333), which may play a key role in the defense against herbivores (Achotegui-Castells et al., 2013), was an enriched molecular function among tissue-specific genes in needles, but also in embryos and vegetative buds. For example, reactive oxygen species (ROS) related biological processes (GO:0006800 and GO:0042743, GO:0034614) and molecular functions (GO:0004601, GO:0004364) were enriched among the GO terms in the tissue-specific genes of embryos, which is consistent with an active ROS protection in developing tissues. In the phloem, a special differentiation process, syncytium formation (GO:0006949), indicating the interconnection of phloem sieve elements to generate a transport route (Geldner, 2014) was an enriched biological process among the tissue specific genes.

Megagametophyte-specific genes have crucial functions in seed germination and energy conversion

Gymnosperms are characterized by the haploid female gametophyte tissue, the megagametophyte, which surrounds the embryo in developing and mature seeds. The megagametophyte can be considered a functional homolog of the endosperm in angiosperms due to its role as a nourishing tissue (King & Gifford, 1997; Costa, Gutièrrez-Marcos & Dickinson, 2004). However, the megagametophyte develops from a haploid megaspore before the fertilization (Singh, 1978) and is therefore entirely maternally inherited unlike the diploid or triploid endosperms of biparental origin (Williams & Friedman, 2002; Baroux, Spillane & Grossniklaus, 2002). To give an example of the potential uses of the dataset, we provide a more detailed description of the megagametophyte expression profile, but leave the in-depth analysis of the other tissues for later investigations.

Among highly expressed and up-regulated genes in the megagametophyte were malate synthase (EC 2.3.3.9) and isocitrate lyase (EC 4.1.3.1) that are essential in glyoxylate cycle converting lipids into carbohydrates in seeds (Ching, 1970), as well as other glyoxysomal proteins like Acetyl-CoA acyltransferase (EC 2.3.1.16), ABC transporter and peroxisomal fatty acid beta-oxidation multifunctional protein AIM1 (Graham, 2008). Seed storage related genes such as 2S seed storage-like protein, 11S globulin seed storage protein 2 and 13S globulin basic chain and some isocitrate lyase copies were completely megagametophyte specific (τ = 1). Antimicrobial and antifungal protein coding genes were the most highly expressed among annotated megagametophyte-upregulated genes.

The enriched GO terms of biological processes and molecular functions in the megagametophyte tissue-specific genes included seed germination and the mobilization of nutrient reserves. Nutrient reservoir activity (GO:0045735) indicated the mobilization of energy sources from the megagametophyte for seed germination and early seedling growth, as well as lipid catabolic processes (e.g. GO:0016042, GO:0044242). Malate dehydrogenase activity (GO:0016615) and heme binding (GO:0020037), which mostly originated from the cytochrome P450 enzymes containing heme cofactors (Xu, Wang & Guo, 2015), reflected the resume of active metabolism. Also, response to ROS (GO:0034614) and antioxidant activity (GO:0016209) suggested active metabolism and signaling. ROS are natural by-products of metabolism and may be detrimental to seed viability because they can cause oxidative stress. However, in the seed ROS also work as signals which underpin the breaking of dormancy and provide protection against pathogens (Jeevan Kumar et al., 2015). Megagametophyte cells showed responses to hormone stimulus (GO:0032870) and the function of hormone-mediated signaling pathways (GO:0009755) including abscisic acid (GO:0009738), auxin (GO:0009734) and ethylene (GO:0009873) which also belong to the molecular networks regulating seed dormancy and germination (Seo et al., 2009; Guangwu & Xuwen, 2014; Miransari & Smith, 2014; Shu et al., 2016). Cellulose biosynthetic process (GO:0030244) and primary cell wall biogenesis (GO:0009833) suggest that cell walls in the megagametophyte may participate in water retention and give mechanical support to the germinating embryo (Otegui, 2007). Similarly to previous findings in P. sylvestris (Merino et al., 2016) megagametophytes, we found enrichment for processes involved in the response to chemical and endogenous stimuli (GO:0042221, GO:0071495 ). Merino et al. (2016) suggested that the megagametophyte could also be involved in the regulation of the embryo development through the induction of signaling pathways triggered by sensing environmental signals in a similar way the angiosperms’ endosperm does (Yan et al., 2014). Altogether, our findings show that the megagametophyte is not just a reserve nutrition for the germinating embryo, but a metabolically active tissue contributing in multiple ways to seed germination and, thus, underline the importance of the haploid stage in P. sylvestris life cycle.

Several enzymes widely used in allozyme-based population genetic studies ((Szmidt & Muona, 1989) and references therein) such as aconitate hydratase (EC 4.2.1.3), malate dehydrogenase (EC 1.1.1.37) and aspartate aminotransferase (EC 2.6.1.1) were megagametophyte-specific and among the top 50 expressed genes in the tissue. As they may be more prone to natural selection against recessive deleterious variants when expressed at the haploid stage, early population genetic analyses may have bias in for example estimates of the overall genetic diversity based on these loci as highly expressed genes are known to be under strong purifying selection.

Conclusions

We provide a widely and interdisciplinary applicable genome-wide atlas of tissue-level transcription patterns based on RNA-seq for economically and ecologically significant coniferous tree P. sylvestris. Quantitative data and analysis of expression level, as well as breadth and tissue specificity are provided for 715,398 different putative genes. The mapping and bioinformatic analyses of gene expression are based on the most complete and high-quality reference transcriptome of P. sylvestris available to date (Ojeda et al., 2019). Previous transcriptome studies of P. sylvestris have concentrated on a narrow set of tissues in each study such as wood (Paasela et al., 2017), embryo (Merino et al., 2016), and needles (Wachowiak et al., 2015; Duarte, Volkova & Geras’kin, 2019) or focused on a limited set of genes (Guseva, Biriukov & Sadovsky, 2020). The present study allows comparison across a wide set of genes expressed in the above-ground parts of adult P. sylvestris trees growing in a natural forest.

In addition to genome sequence annotations, we foresee multiple potential uses for the dataset. Level and breadth of gene expression are known to be linked to the evolutionary rate and level of conservation (Lemos et al., 2005, Brawand et al., 2011, Yang & Wang, 2013). By combining our data with similar data in other conifers or angiosperms it is possible to study the evolutionary conservation of expression patterns, or the differences in evolutionary rates across tissue-specific expression levels and gain a deeper understanding of the determinants and main factors affecting for example rate of adaptive evolution and dynamics at the genome level. The response of trees to a combination of different stresses is unique and cannot be directly extrapolated from studying only single stressors in experimental conditions (Niinemets, 2010). The transcriptome resource for adult P. sylvestris trees growing under natural conditions, where they are simultaneously exposed to a number of different abiotic and biotic stresses as well as interactions with other organisms, provides a valuable tool also for physiological studies. Finally, un-annotated conifer genes with high expression or tissue specificity can open up whole new research avenues, independent of the previously available knowledge based on angiosperm model plants such as A. thaliana and Populus.

Supplementary information

All the supplementary files can be found at DOI 10.6084/m9.figshare.c.5264255.v1.

Additional Information and Declarations

Competing Interests

Author Contributions

Data Availability

The authors declare that they have no competing interests.

Sandra Cervantes conceived and designed the experiments, performed the experiments, analyzed the data, prepared figures and/or tables, authored or reviewed drafts of the paper, and approved the final draft.

Jaana Vuosku analyzed the data, authored or reviewed drafts of the paper, and approved the final draft.

Tanja Pyhäjärvi conceived and designed the experiments, performed the experiments, analyzed the data, prepared figures and/or tables, authored or reviewed drafts of the paper, and approved the final draft.

The following information was supplied regarding data availability:

Clean reads corresponding to the 30 samples (six genotypes × five tissues) used in the transcript quantification are available at NCBI GenBank: PRJNA531617.

The data and tables are available at Figshare: Cervantes, Sandra; Vuosku, Jaana; Pyhäjärvi, Tanja (2021): Atlas of tissue-specific and tissue-preferential gene expression in ecologically and economically significant conifer Pinus sylvestris - Associated data and tables. figshare. Collection. https://doi.org/10.6084/m9.figshare.c.5264255.v1.

All code used is freely available at the following sites:

- Transcript quantification and abundance matrices construction:

https://github.com/trinityrnaseq/trinityrnaseq/wiki/Trinity-Transcript-Quantification.

- Quality control: https://github.com/trinityrnaseq/trinityrnaseq/wiki/QC-Samples-and-Biological-Replicates.

- Differential expression analysis and identification of tissue preferentially upregulated genes: https://github.com/trinityrnaseq/trinityrnaseq/wiki/Trinity-Differential-Expression; https://github.com/trinityrnaseq/trinityrnaseq/wiki/Sample-Specific-Expression.

- Tissue-specific expression: https://github.com/severinEvo/gene_expression/blob/master/tau.R.

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
