# Peer review of "Atlas of tissue-specific and tissue-preferential gene expression in ecologically and economically significant conifer Pinus sylvestris"

_PeerJ, doi:10.7717/peerj.11781_

## Round 0.1 · original submission · Major Revisions

Dear author,

As you can see from the comments of our reviewers, a major revision is required. Be aware that we will send your revision to the original reviewers.

Kind regards
Michael Wink
Academic Editor

Reviewer 1 ·

Basic reporting

The authors of the manuscript entitle “Atlas of tissue-specific and tissue-preferential gene expression in ecologically and economically significant conifer Pinus sylvestris” have analyzed the transcriptome of different organs/tissues from the Scot pine by RNA-seq determining the profiles and preferential gene expression in the different organs/tissues.
The manuscript is well written although there with several minor grammatical or form mistakes. For example:
Line 53: Change “as for example Pinus and Picea” to “as for example Pinus and Picea genres”.
Line 80: Change “Canas et al. 2017” to “Cañas et al. 2017”.
Lines 98-108: Change letter color to black.
Line 143: Change (Ojeda et al. 2019) from cursive to normal case.
Literature cited cover the subject of the manuscript and the structure, figures and supplemental data are correct.

Experimental design

The research is original, but the relevance is limited since transcriptomic studies now are easy to make and accessible for near every research group working on conifers. It is true that the variety of organs and tissues, and development and environmental conditions for this kind of plants is very high. For this reason, it is necessary to increase the transcriptomic data in coniferous trees, but the selected group of samples is limited and without a clear objective.
Methods are in general well described and the statistical approach is complex, more advanced than a simple K-means or a correlation clustering. However, I have some comments about some methodological approaches.
Detailed comments:
Line 154: Please describe the initial quantification and quality controls of total RNA. Before to make polyA isolation from total RNA to make sequencing libraries the integrity of the RNA must be OK. For example, using a Bioanalyzer the RIN must be higher than 6.5-7.
Line 170: Please, describe the trimming process.
Line 171: Please, improve the sentence. It is difficult to follow, the reference must be in the journal format (not with a number). I have a problem with the assembly method. If you want to analyse for the first time the transcriptome of different tissues/organs it is not recommended to use a guided alignment with a previous transcriptome as reference (with a genome is OK). In this sense, you will loss new transcripts (splicing variants and transcripts for not identified genes) expressed in your samples. See in the line 282 that only the 76% of the reads are mapped in the reference.
Lines 181-187: The contaminant removal process is very complicated. Why have you removed contaminants from contigs and not from reads? There are different software that remove contaminant reads.
The number of contigs is very huge (1.2 millions). You must consider to use CD-HIT to reduce the sequence redundancy.
I think it will be necessary to use REVIGO tool to highlight the more important and non-redundant GO terms. Additionally, I miss RT-qPCR validation of the results.

Validity of the findings

Although new transcriptome data in conifers always is interesting, I think that in the present manuscript the main findings have been lost since de novo assembly has been made through a reference-guided approach. Additionally, an annotation based on a previous reference transcriptome limits the originality of the work. These are important concerns since a database containing transcriptome and annotation data (GymnoPlaza) from Pinus sylvestris is accessible for researchers

Additional comments

I have a concern about the use of the word “tissue” in the manuscript including the title. It is true that this term is clarified in the Discussion section. But I think this must be done before indicating that some of your samples are organs and not tissues, maybe in the Introduction.

Reviewer 2 ·

Basic reporting

The article is clear and unambigous, English is professional.

Wide background, well cited.

Figures and data are clear.

The hypotheses that tissue specific genes differentially expressed could be detected was confirmed. The megagametophyte was not the tissues with most tissue specific genes as hypothesized, rather needles.

Experimental design

The research contributes original new data, as previously fewer tissues had been sampled from a single stage or condition.

It completes previous studies, it provides a support for genome sequencing.

Yes, although I have some comments/questions below.

Standard methods were used for tissue collection and data analyses- but see comments.

Validity of the findings

Yes. Extensive and relevant results and discussions sections.


Stastistics are suitable.

Additional comments

Dear author,

The 'Atlas of tissue-specific and tissue-preferential gene expression in ecologically and economically significant conifer Pinus sylvestris' is an important contribution towards the genome sequencing effort of this tree.

I have a few comments/questions:

For the sampling: 6 non related individuals were sampled in the field, is there a reason for not using clones, which provide more statistical power for the reconstruction of the transcripts?

For the analysis first with Salmon (pseudo-alignment) and then Trinity, how did these two methods compare? I did not find it in the results-discussion, was this useful, did they correlate well with the number of transcripts?

Is there a reason for the authors not to use a single copy orthologous genes analysis to get an estimate of transcriptome completeness, like BUSCO?
This might provide additional relevant information.

Suggestions:

P. 14 line 213: please explain here PUR again, as it had only been explained in the introduction (and many readers might overlook it)

P. 28, lines 514-516: Might be useful to indicate which type of bias: i.e. underestimating genetic variation

Best wishes!

---

## Round 0.2 · accepted · Accept

Dear authors
thank you for your revision which meets the recommendations of the reviewers. Therefore, we can accept your manuscript.
Congratulations
Best Regards
Michael Wink
AE

Reviewer 1 ·

Basic reporting

After authors' reponse I have no additional comments. The authors have appropiately answered to my comments and revised the manuscript.

Experimental design

I only have a minor comment or opinion, not for a new round of revision. There is a generalized opinion that RT-qPCR validation is not necessary for RNA-seq experiments. It is true that RNA-seq is very robust technique but the analyses depend on reference sequences used for mapping. In the case of well stablished model such as human or Arabidopsis with extremely well annotated genomes this can be, they do not need RT-qPCR validation experiments (although I consider that is necessary to determine if the sample files have been used in the proper order and not inverted). But the RNA-seq analyses based on transcriptomes and assemblies this is different. You can find splicing isoforms, misassembled contigs, etc. In fact, the library synthesis can introduce some biases. In many cases Illumina results do not well correlate with PacBio or ONT sequencing of unamplified samples, in an intuitive manner it is easy to think that results of the 3rd generation NGS sequencing in this case are more closed to reality.

Validity of the findings

After authors' reponse I have no additional comments. The authors have appropiately answered to my comments and revised the manuscript.

Additional comments

After authors' reponse I have no additional comments. The authors have appropiately answered to my comments and revised the manuscript.

Reviewer 2 ·

Basic reporting

Very professional sharing of the document via Google Docs, but please make sure that the reviewers can only read and not edit, as information can be wrongly deleted.

Else no comment.

Experimental design

Many ambiguities were cleared up.
I don't find sharing the contaminant list (https://figshare.com/articles/dataset/Data_S1_List_of_putative_contaminant_contigs_removed_from_quantification_files_txt/13121849) very useful. Please share the sequences (fasta) as well.

Validity of the findings

No comment.

Additional comments

Dear authors,

Thank you for addressing my concerns, there is a very significant improvement of the manuscript.

Best wishes!